# A voxel-based analysis of cerebral blood flow abnormalities in obsessive-compulsive disorder using pseudo-continuous arterial spin labeling MRI

Daichi Momosaka[1]*, Osamu Togao[1], Akio Hiwatashi[2], Koji Yamashita[1], Kazufumi Kikuchi[1], Hirofumi Tomiyama[3], Tomohiro Nakao[3], Keitaro Murayama[3], Yuriko Suzuki[4], Hiroshi Honda[1]

1 Department of Clinical Radiology, Graduate School of Medical Sciences, Kyushu University, Fukuoka, Japan, 2 Department of Molecular Imaging & Diagnosis, Graduate School of Medical Sciences, Kyushu University, Fukuoka, Japan, 3 Department of Neuropsychiatry, Graduate School of Medical Sciences, Kyushu University, Fukuoka, Japan, 4 Institute of Biomedical Engineering, University of Oxford, Oxford, United Kingdom

* dmomosaka@icloud.com

**Data Availability Statement:** All relevant data are within the manuscript and its Supporting Information files.

## Abstract

### Objective

To identify abnormalities of regional cerebral blood flow (rCBF) in individuals with obsessive-compulsive disorder (OCD) by conducting a voxel-based analysis of pseudo-continuous arterial spin labeling (pCASL) perfusion images.

### Materials and methods

This prospective study included 23 OCD patients (nine males, 14 females; age 21–62 years; mean ± SD 37.2 ± 10.7 years) diagnosed based on DSM-IV-TR criteria and 64 healthy controls (27 males, 37 females; age 20–64 years; mean ± SD 38.3 ± 12.8 years). Subjects were recruited from October 2011 to August 2017. Imaging was performed on a 3T scanner. Quantitative rCBF maps generated from pCASL images were co-registered and resliced with the three-dimensional T1-weighted images, and then spatially normalized to a brain template and smoothed. We used statistical nonparametric mapping to assess the differences in rCBF and gray matter volume between the OCD and control groups. The significance level was set at the p-value <0.05 with family-wise error rate correction for multiple comparisons.

### Results

Compared to the control group, there were significant rCBF reductions in the right putamen, right frontal operculum, left midcingulate cortex, and right temporal pole in the OCD group. There were no significant between-group differences in the gray matter volume.

**Funding:** This work was supported by JSPS KAKENHI grants, no. JP17K10410 and no. JP20K08111 (https://www.jsps.go.jp). The funders had no role in study design, data collection and analysis, decision to publish, or preparation of the manuscript.

**Competing interests:** We have declared that no competing interests exist.

**Abbreviations:** ASL, arterial spin labeling; CSTC, cortico-striatal-thalamic circuits; OCD, obsessive-compulsive disorder; pCASL, pseudo-continuous arterial spin labeling; rCBF, regional cerebral blood flow; ROI, region of interest; SPM, statistical parametric mapping; VBM, voxel-based morphometry; Y-BOCS, Yale-Brown Obsessive Compulsive Scale.

## Conclusion

The pCASL imaging noninvasively detected physiologically disrupted areas without structural abnormalities in OCD patients. The rCBF reductions observed in these regions in OCD patients could be associated with the pathophysiology of OCD.

## Introduction

Obsessive-compulsive disorder (OCD) is a common mental disorder that has a lifetime prevalence rate of 2%–3% in the general population [1]. OCD is characterized by recurrent intrusive thoughts and repetitive, ritualistic behaviors that negatively affect the affected person's daily life [2]. It is now widely accepted that abnormalities of the cortico-striatal-thalamic circuits (CSTC), particularly involving the orbitofrontal cortex, anterior cingulate cortex, thalamus and striatum, play an important role in the pathophysiology of OCD [3]. It was also reported that the underlying pathology is not limited to the CSTC but also involves abnormalities in additional brain systems, particularly including dorsolateral frontal and parietal regions [3].

Although studies using positron emission tomography (PET) or single photon emission tomography (SPECT) have been conducted to examine abnormalities of regional cerebral blood flow (rCBF) in individuals with OCD [4–13], the majority of these studies assessed rCBF by using regions of interest (ROI), which might generally lack efficient reproducibility and reliability [14]. Moreover, the ROIs used in the studies tended to lump together heterogeneous subregions with separate functional roles. With ROI-based methods, only selected portions of the brain—rather than the entire brain—are examined [7]. A voxel-based analysis has several advantages over the ROI-based methods [15]. It is an objective and automated method that can be used to investigate the functional parameters, including rCBF, of the entire brain, without the need to define anatomical boundaries.

Radioisotope examinations pose the risk of exposure to ionizing radiation. In contrast, arterial spin labeling (ASL) MRI is a noninvasive technique that can measure cerebral perfusion without radiation exposure or the use of contrast agents [16]. We conducted the present study to determine the differences in rCBF between patients with OCD and healthy controls by assessing the results obtained with pseudo-continuous ASL (pCASL) perfusion imaging. This modality combines the advantages of pulsed ASL and continuous ASL; pulsed ASL has relative insensitivity to magnetization transfer-related artifacts and a lower specific absorption rate, whereas continuous ASL has a higher signal-to-noise ratio (SNR) [16]. ASL perfusion imaging has been applied to several psychiatric diseases [17, 18], but there have been no studies that used ASL to investigate perfusion abnormalities in OCD patients.

Based on the results of prior studies, we hypothesized that the regional cerebral perfusion would differ between patients with OCD and healthy controls in regions in the CSTC, especially for the orbitofrontal cortex, anterior cingulate cortex, thalamus and striatum. The purpose of the present study was to identify abnormalities of rCBF in the CSTC as well as in other regions in OCD patients by performing a voxel-based analysis of pCASL perfusion images.

## Materials and methods

### Study population

This prospective observational study was approved by the Kyushu University Institutional Review Board for Clinical Research, and all recruited subjects signed an informed consent

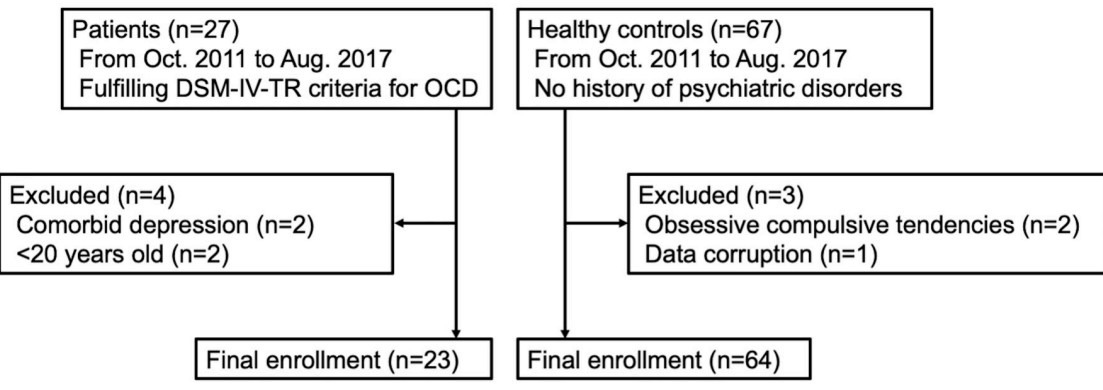

**Fig 1. Study population flowchart.**

form before the study. Patients were recruited from the department of neuropsychiatry at our institution from October 2011 to August 2017. Patients fulfilling the Diagnostic and Statistical Manual of Mental Disorders, Fourth Edition (DSM-IV) criteria for OCD based on the structured clinical interview for DSM-IV Axis I Disorders–Patient edition were selected [2].

The exclusion criteria were: (a) age <20 or >65 years; (b) pregnant; (c) current/past alcohol or drug abuse; (d) comorbid Axis I diagnosis on the above-mentioned structured clinical interview; (e) neurological or other serious physical illness; and (f) contraindications to MRI. All of the patients were medication-free for ≥4 weeks before the study. The severity of each patient's current symptoms of OCD was measured by the Yale-Brown Obsessive Compulsive Scale (Y-BOCS) [19]. Healthy controls were recruited from the nearby communities. A group of healthy controls who met the exclusion criteria described above with no history of psychiatric disorders were selected. Healthy subjects who showed any obsessive-compulsive tendencies (Y-BOCS score >0) were excluded. Fig 1 presents a summary flowchart of the subject selection and Table 1 summarizes the demographic and clinical characteristics of the patients and healthy controls. The detailed information of the OCD patients and healthy controls are provided in S1 and S2 Appendices.

## MRI data acquisition

Imaging was performed on a 3T MR system (Philips Healthcare, Best, The Netherlands) with an eight-channel head coil. High-spatial-resolution 3D T1-weighted images were acquired with a turbo field echo sequence in the sagittal plane with the following parameters: repetition

**Table 1. Demographic and clinical characteristics of the subjects.**

|  | OCD (n = 23) | Healthy controls (n = 64) | p-value |
|---|---|---|---|
| Gender, male:female | 9: 14 | 27: 37 | 0.7981 |
| Age, yrs | 37.2 ± 10.7 | 38.3 ± 12.8 | 0.7265 |
| Education, yrs | 12.8 ± 2.6 | 15.5 ± 1.8 | <0.0001* |
| Total brain volume, L | 1.15 ± 0.11 | 1.14 ± 0.1 | 0.4949 |
| Illness duration, yrs | 14.3 ± 12.3 |  |  |
| Y-BOCS | 22.6 ± 6.5 |  |  |

Data are mean ± SD.

*p <0.05. OCD, obsessive-compulsive disorder; Y-BOCS, Yale-Brown Obsessive-Compulsive Scale.

time/echo time, 8.21/3.78 msec; inversion time, 1026 msec; flip angle, 8˚; effective section thickness, 1.0 mm; slab thickness, 190 mm; matrix, 240 × 240; field of view, 240 × 240 mm; number of signals acquired, 1; scan duration, 5 min 23 sec. All scans resulted in 190 contiguous slices through the brain.

The imaging parameters for the pCASL experiments were as follows: single-shot gradient-echo echo planar imaging in combination with parallel imaging (sensitivity encoding factor 2.0); repetition time/echo time, 4200/8.56 msec; matrix, 64 × 64; field of view, 240 × 240 mm; in-plane resolution, 3.75 × 3.75 mm; 20 slices acquired in ascending order; slice thickness, 6 mm; slice gap, 1 mm; labeling duration, 1650 msec; post-labeling delay, 1525 msec. Forty pairs of control/label images were acquired and then averaged. The total scan duration was 5 min 44 sec. The echo planar imaging M0 images were separately obtained with the same geometry and imaging parameters as the pCASL without labeling.

## Preprocessing of the data

**rCBF measurements with pCASL.** The quantitative rCBF maps were calculated and generated from raw pCASL images and M0 images with an in-house MATLAB program using the following equation [20]:

$$rCBF = \frac{6000 \cdot \lambda \cdot (SI_{control} - SI_{label}) \cdot e^{\frac{PLD}{T1_{blood}}}}{2 \cdot \alpha \cdot T1_{blood} \cdot SI_{PD} \cdot \left(1 - e^{-\frac{\tau}{T1_{blood}}}\right)} \, [ml/min/100g],$$

Where $\lambda$ is the brain/blood partition coefficient in ml/g, $SI_{control}$ and $SI_{label}$ are the time-averaged signal intensities in the control and label images, respectively, $T1_{blood}$ is the longitudinal relaxation time of blood in seconds, $\alpha$ is the labeling efficiency, $SI_{PD}$ is the signal intensity of a proton density-weighted image, $\tau$ is the label duration, and PLD is the post-labeling delay. The parameters used in the present study were: $\lambda$ = 0.9 (assumed), $T1_{blood}$ = 1650 msec (assumed), $\alpha$ = 0.85 (assumed), $\tau$ = 1650 msec (calculated) and PLD = 1525 msec (calculated). The $SI_{PD}$ was derived from the echo planar imaging M0 images.

**Preprocessing of the rCBF maps.** The conversion from the Digital Imaging and Communications in Medicine (DICOM) format to the NifTI-1 format, the preprocessing, and the quality check of the acquired images were performed with Statistical Parametric Mapping (SPM12) software (Functional Imaging Laboratory, Wellcome Trust Centre for Neuroimaging, Institute of Neurology at University College London, UK) running on MATLAB R2016a (MathWorks Inc., Sherborn, MA).

Fig 2 illustrates the following preprocessing steps of the rCBF maps. The generated quantitative raw rCBF maps were co-registered and resliced with the 3D T1-weighted images. This was done by first co-registering and reslicing the M0 images to the 3D T1-weighted images, and then applying the same co-registration and reslicing to the raw rCBF maps. Because M0 images retain more anatomical information than the raw rCBF maps, this process enables a more accurate co-registration of the raw rCBF maps to the 3D T1-weighted images. Next, the 3D T1-weighted images were spatially normalized to the International Consortium for Brain Mapping (ICBM) template for East Asian brains [21]; then, the same spatial transformation was applied to the rCBF maps, thus allowing a voxel-wise analysis of the rCBF maps in a common stereotaxic space. Because 3D T1-weighted images have more anatomical information than rCBF maps, this process enables a more accurate normalization of rCBF maps to the brain template. Finally, the rCBF maps were smoothed with a 12-mm full width at half maximum Gaussian kernel in order to decrease spatial noise and compensate for the inexactitude of normalization.

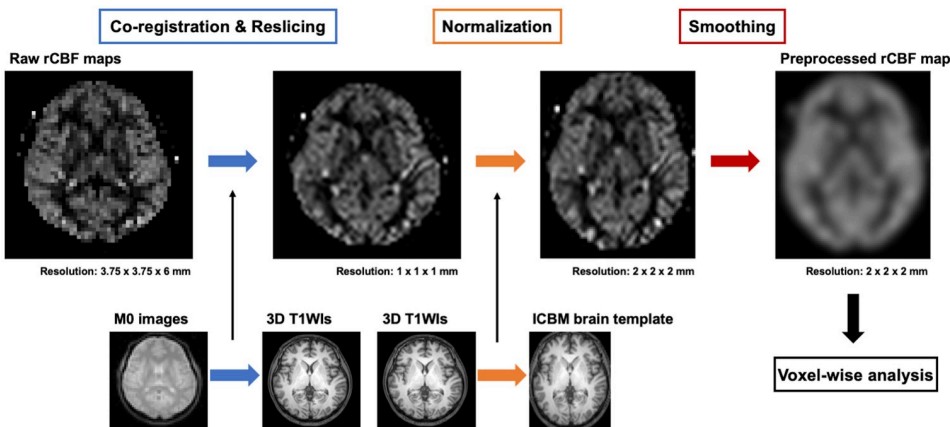

**Fig 2. Preprocessing steps of the rCBF maps.**

**Preprocessing of Voxel-Based Morphometry (VBM).** For the preprocessing of the results of the voxel-based morphometry (VBM), first, the gray matter was segmented from 3D T1-weighted images with the use of the SPM12 program. Next, the subject's gray matter was co-registered and normalized using the diffeomorphic anatomical registration through exponentiated lie algebra (DARTEL) technique by using SPM12 software running on MATLAB R2016a. The details of the image preprocessing of VBM are described elsewhere [22].

## Statistical analysis

The statistical analyses were performed using statistical nonparametric mapping (SnPM13) software (http://warwick.ac.uk/snpm) [23], which is a toolbox of SPM12 software running on MATLAB R2016a. The nonparametric permutation approach is preferable for experimental designs implying low degrees of freedom, including small sample size problems such as between-group analyses involving small numbers of subjects [23]. We compared the rCBF and gray matter volume of the OCD patients and healthy controls by running 10,000 permutations, using the subjects' age, gender, and education years as nuisance covariates. An explicit mask was constructed from the average smoothed image of all of the subjects, using the SPM Masking Toolbox (http://www0.cs.ucl.ac.uk/staff/g.ridgway/masking/) [24] to restrict the statistical analyses to voxels that represent gray matter. The total brain volumes of the subjects were used as global values in the global calculation. The other parameters of the analysis were as follows: variance smoothing, 0 0 0; memory usage, high; cluster inference, none; threshold masking, none; implicit mask, yes; overall grand mean scaling, No; normalization, ANCOVA. Regression analyses investigating the relationship between the rCBF and the Y-BOCS score or the illness duration within the OCD group were performed on SnPM13 using the same parameters described above. P-values <0.05 were considered significant, with the family-wise error rate correction for multiple comparisons. Clusters smaller than 50 voxels or located outside of the brain were excluded from the subsequent analyses.

## Results

Compared to the healthy controls, the patients with OCD showed significant rCBF reductions in the right putamen, the right frontal operculum, the left midcingulate cortex, and the right temporal pole (Fig 3, Table 2). S3 Appendix presents the result of the ANCOVA with SnPM13 software. There were no areas with a significant rCBF elevation in the OCD patients compared

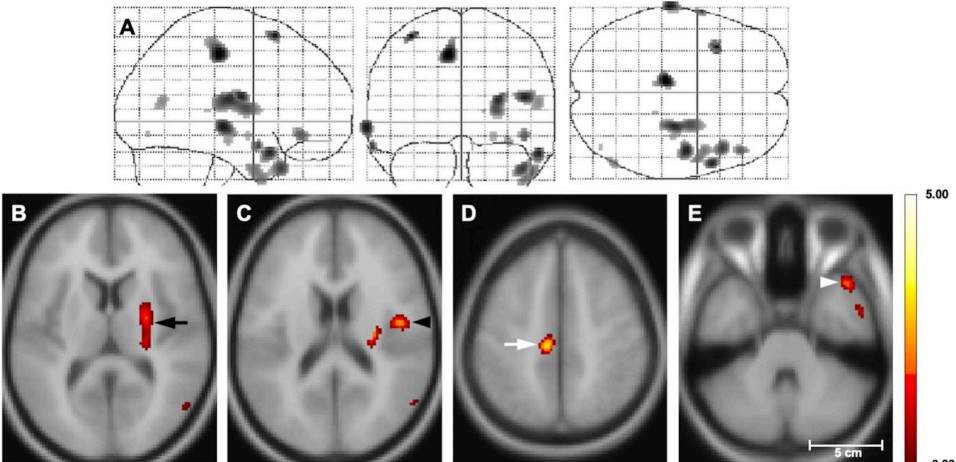

**Fig 3. Statistical nonparametric T map of rCBF reduction in the OCD patients vs. healthy controls.** (A) The three orthogonal planes represent a maximum intensity projection (MIP) 'glass brain'. (B–D) Selected planes show results superimposed on MNI152 T1 template. Color bar: Student's T-values. Significant reductions of rCBF are observed in the right putamen (black arrow), right frontal operculum (black arrowhead), left midcingulate cortex (white arrow), and right temporal pole (white arrowhead) in the OCD patients compared to the controls (thresholded at T > 3.93, p<0.05 with the family-wise error rate correction for multiple comparisons in the SnPM analysis). There were no areas of increased rCBF in the OCD patients (data not shown). rCBF, regional cerebral blood flow; OCD, obsessive-compulsive disorder; SnPM, statistical nonparametric mapping.

to the controls. The regression analyses revealed no significant correlations between the rCBF and Y-BOCS score or illness duration in the OCD group.

There was no significant difference in the regional gray matter volume between the OCD patients and healthy controls.

We performed an additional ROI-based analysis for the areas that showed decreased rCBF in the voxel-based analysis described above, in order to compare the two analytical methods (for details, see S4 Appendix). The results of the ROI-based analysis demonstrated that the mean rCBF values within the right putamen and the right temporal pole were significantly lower in the OCD patients compared to the healthy controls; there were no significant differences in mean rCBF in the left midcingulate cortex or right frontal operculum between the groups.

## Discussion

We investigated rCBF abnormalities in drug-free OCD patients in a comparison with healthy controls by using pCASL perfusion MRI and a voxel-based image analysis. The results

**Table 2. Regions showing decreased rCBF in the OCD patients compared to the healthy controls.**

|  | Coordinates, mm | | | Size, voxels | T-value |
|---|---|---|---|---|---|
|  | *x* | *y* | *z* |  |  |
| R. putamen | 26 | −18 | 16 | 301 | 4.40 |
| R. frontal operculum | 46 | −8 | 18 | 92 | 4.44 |
| L. midcingulate cortex | −10 | −24 | 48 | 160 | 4.59 |
| R. temporal pole | 44 | 20 | −34 | 176 | 4.36 |

voxel size = 2 × 2 × 2 mm³. R, right; L, left; rCBF, regional cerebral blood flow; OCD, obsessive-compulsive disorder.

demonstrated reduced rCBF in the right putamen, right frontal operculum, left midcingulate cortex, and right temporal pole in the OCD patients.

Many studies using PET or SPECT have been undertaken to assess rCBF changes in OCD (S5 Appendix) [4–13], and although these studies vary regarding the number of subjects, neuroimaging modality, and analytic method, rCBF abnormalities have consistently been identified in the CSTC of OCD patients, including the orbitofrontal cortex [4, 7, 9, 12], the anterior cingulate cortex [5–7, 11, 12], the caudate nucleus [6, 10, 11], and the thalamus [6, 8, 9]. Our present findings are partially consistent with these reports in terms of the rCBF abnormality in the putamen observed in the OCD patients, but we observed no significant abnormalities in the other areas of the CSTC.

It is widely accepted that regional perfusion is significantly correlated with regional neuronal activity (neurovascular coupling) [25], and the rCBF reduction observed in the present OCD patients may therefore reflect a functional decline of these areas.

The putamen is a part of the CSTC, and it is involved in planning and execution [26]. A meta-analysis of functional MRI studies reported that OCD patients showed lower activation of the putamen during executive functioning compared to healthy controls [27]. The hypoperfusion in the putamen that we observed herein is in line with this result, considering the theory of neurovascular coupling. An ROI-based SPECT study revealed decreased rCBF in the basal ganglia of OCD patients [6]. On the other hand, other ROI-based SPECT studies focusing on adult OCD patients on medication [10] or pediatric OCD patients [11] reported increased rCBF in their patients' basal ganglia. We speculate that factors such as the differences in analytical method (ROI-based or voxel-based approach), medication use, and/or the patients' age could be causes of these discrepant observations. It appears that hypoperfusion in the putamen may underlie the disturbance of executive function observed in OCD.

In the present series, the OCD patients also showed foci of reduced rCBF in the medial frontal gyrus, encompassing the midcingulate cortex. The medial frontal gyrus was reported to be involved in the cognitive regulation of emotional behavior including fear and anxiety [28], which are core symptoms of OCD [2]. An extensive review of studies of cingulate neurobiology emphasized the importance of checking symptomatology in relation to the midcingulate cortex impairment [29]. A functional MRI study with symptom-provocation protocols suggested that hyperactivation in the midcingulate cortex was associated with checking provocation [30]. Regarding the rCBF at resting state, a voxel-based SPECT study described significantly lower rCBF values in the midcingulate cortex in patients with early-onset OCD compared to those of healthy controls [31]. Considering the above-described findings, it seems likely that the functional disturbance of the midcingulate cortex is involved in some aspects of the pathophysiology of OCD. In addition, neuronal activity or perfusion of the midcingulate cortex might dynamically change in association with checking symptoms in OCD patients.

The OCD patients in our present series also showed a focus of decreased rCBF in the frontal operculum compared to the controls. The frontal operculum correlates with resistance to compulsions in OCD patients [32]. A functional MRI study reported that OCD patients showed lower spontaneous brain activity in the Rolandic operculum compared to healthy controls [33]. Although there are no reports describing rCBF changes in this area, reduced gray matter volume was reported in a VBM study, and increased mean diffusivity in this region was also demonstrated by a diffusion tensor imaging study [34]. The decreased rCBF in the frontal operculum might be associated with the decline of resistance to compulsions and with the morphological and microstructural changes observed in these studies.

We observed a significant rCBF reduction in the right temporal pole in the present OCD patients. A functional MRI investigation demonstrated that the temporal pole exhibits dense functional connections to the CSTC and that it is a part of a control mechanism that down-

regulates emotional saliency during the processing of emotional stimuli in healthy subjects [35]. In addition, OCD patients had significantly decreased functional connectivity between the temporal pole and the orbitofrontal cortex, which is a part of the CSTC [36]. Based on these results, we suggest that it is likely that decreased functioning of the temporal pole may cause an impaired down-regulation of emotional stimuli to the CSTC, which may be associated with the symptomatology of OCD.

In evaluations using diffusion tensor imaging (DTI), increased fractional anisotropy in the putamen [37] and increased mean diffusivity in the frontal operculum [34] were observed. These changes of the DTI parameters might reflect alterations of the underlying white matter microstructure, including fiber packing, fiber diameter, the thickness of the myelin sheaths, and the directionality of the fibers [38]. A composite study using ASL and DTI demonstrated a significant relationship between rCBF and the subcortical white-matter integrity assessed by fractional anisotropy and the diffusivity in healthy subjects [39]. The reductions of rCBF in the putamen and the operculum that we observed herein may result from the microstructural disturbance in these regions.

In the present VBM analysis, unlike many other studies, there were no significant differences in the gray matter volume between the OCD patients and healthy controls. This discrepancy may be due to the limited sample size (23 OCD patients, 64 healthy controls). Previous VBM studies with larger sample sizes reported that the changes in gray matter volume were found in the areas that are similar or adjacent to those in which the present ASL analysis showed rCBF reduction, i.e., the medial prefrontal cortex, cingulate cortex, and insula [40, 41]. Considering these findings, we speculate that the ASL could sensitively detect physiologically impaired areas without structural abnormalities even in the present small sample size.

We also observed that there was no area where the rCBF was significantly correlated with the Y-BOCS score, whereas other investigations demonstrated positive correlations between symptom severity and the rCBF in the CSTC [7, 9, 10]. This discrepancy indicates that cerebral perfusion changes do not invariably correlate with symptom severity in OCD. Although it has been reported that the duration of illness affects the volume of the striatal areas, medial frontal, orbitofrontal, and insulo-opercular areas in patients with OCD [42], we observed no significant correlations between rCBF and the illness duration in the present study. The negative results for Y-BOCS and illness duration shown herein may be due to the highly heterogeneous nature of OCD [43].

Two analytical methods have been used to assess rCBF abnormalities in OCD patients in prior investigations: an ROI-based approach or a voxel-based approach [4–13]. We performed an ROI-based analysis in addition to the main voxel-based analysis in order to compare the two analytical methods, and the results demonstrated that the ROI-based analysis failed to detect rCBF decreases in the right frontal operculum and the left midcingulate cortex. This might have occurred because the ROIs lumped together large heterogeneous subregions with separate functional roles, with the results that small foci of decreased rCBF were missed. It therefore appears that the voxel-based approach has an advantage over the ROI-based approach in detecting smaller areas with rCBF abnormality.

The absence of significant findings involving other areas of the CSTC in the present patients was unexpected, since many studies have reported the presence of rCBF alterations in the CSTC [4–13]. It is possible that the discrepancy between our findings and previous findings is related to unknown differences in clinical characteristics of the patients studied.

Our study has several limitations, the first of which is the small sample size. Second, there was a significant difference in the years of education between the OCD and control groups. It was unclear whether the observed differences persisted and affected the results even after adjustment for years of education. Third, we used the PLD of 1525 msec for the pCASL, and

this value is shorter compared to the recommendation in the consensus paper published by the MR perfusion community [20], i.e., the PLD of 1800 msec for adults <70 years old. However, since our present subjects were relatively young (OCD: mean 37.2 yrs, controls: mean 38.3 yrs), we believe that the PLD of 1525 msec was not too short. Fourth, ASL using echo planar imaging inevitably suffers from susceptibility artifacts (particularly in mesial temporal regions) due to the proximity of the skull base and air-containing structures [44]. Distortions of pCASL images in these regions might have led to a low SNR to some extent; however, as we focused on the between-group differences in quantitative rCBF maps rather than conducting a direct analysis of raw pCASL images, we think that the SNR of the raw pCASL images did not critically affect the final results. Last, a DTI analysis was not performed. For the further clarification of the pathophysiology of OCD from various perspectives, a composite study including ASL, VBM, and DTI analyses is needed.

## Conclusion

Our findings demonstrated that ASL perfusion imaging could noninvasively detect physiologically impaired areas without structural abnormalities in OCD patients. Our pCASL-based study using a voxel-based analysis revealed hypoperfusion in the right putamen, right frontal operculum, and left midcingulate cortex of OCD patients. The rCBF reductions observed in these regions may be associated with the pathophysiology of OCD.

## Supporting information

**S1 Fig.**
(TIFF)

**S1 Appendix. The detailed information of the OCD patients.**
(XLSX)

**S2 Appendix. The detailed information of healthy controls.**
(XLSX)

**S3 Appendix. The result of the ANCOVA with SnPM13 software**
(TIF)

**S4 Appendix. The detailed information of the additional ROI-based analysis.**
(DOCX)

**S5 Appendix. Previous studies showing rCBF changes in OCD patients.**
(XLSX)

## Author Contributions

**Conceptualization:** Daichi Momosaka, Osamu Togao, Koji Yamashita.

**Data curation:** Daichi Momosaka, Osamu Togao.

**Formal analysis:** Daichi Momosaka, Osamu Togao.

**Investigation:** Daichi Momosaka, Osamu Togao, Hirofumi Tomiyama.

**Methodology:** Daichi Momosaka, Osamu Togao, Yuriko Suzuki.

**Project administration:** Daichi Momosaka, Osamu Togao, Hirofumi Tomiyama.

**Resources:** Daichi Momosaka.

**Software:** Yuriko Suzuki.

**Supervision:** Osamu Togao, Koji Yamashita, Kazufumi Kikuchi, Hirofumi Tomiyama, Tomohiro Nakao, Keitaro Murayama, Yuriko Suzuki, Hiroshi Honda.

**Validation:** Daichi Momosaka.

**Visualization:** Daichi Momosaka.

**Writing – original draft:** Daichi Momosaka.

**Writing – review & editing:** Daichi Momosaka, Osamu Togao, Akio Hiwatashi, Koji Yamashita, Kazufumi Kikuchi, Hirofumi Tomiyama, Tomohiro Nakao, Keitaro Murayama.

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
