## [Decision Letter · Decision Letter 0]

6 May 2020

PONE-D-20-05758

A voxel-based analysis of cerebral blood flow abnormalities in obsessive-compulsive disorder using pseudo-continuous arterial spin labeling MRI A cerebral blood flow analysis of obsessive-compulsive disorder using MRI

PLOS ONE

Dear Dr. Momosaka,

Thank you for submitting your manuscript to PLOS ONE. After careful consideration, we feel that it has merit but does not fully meet PLOS ONE’s publication criteria as it currently stands. Therefore, we invite you to submit a revised version of the manuscript that addresses the points raised during the review process.

We would appreciate receiving your revised manuscript by Jun 20 2020 11:59PM. To enhance the reproducibility of your results, we recommend that if applicable you deposit your laboratory protocols in protocols.io, where a protocol can be assigned its own identifier (DOI) such that it can be cited independently in the future. For instructions see: http://journals.plos.org/plosone/s/submission-guidelines#loc-laboratory-protocols

We look forward to receiving your revised manuscript.

Kind regards,

Xi Chen

Academic Editor

PLOS ONE

Journal Requirements:

2. Please amend either the title on the online submission form (via Edit Submission) or the title in the manuscript so that they are identical.

Reviewers' comments:

Reviewer's Responses to Questions

**Comments to the Author**

1. Is the manuscript technically sound, and do the data support the conclusions?

Reviewer #1: Yes

Reviewer #2: Yes

2. Has the statistical analysis been performed appropriately and rigorously? 

Reviewer #1: Yes

Reviewer #2: Yes

3. Have the authors made all data underlying the findings in their manuscript fully available?

Reviewer #1: Yes

Reviewer #2: No

4. Is the manuscript presented in an intelligible fashion and written in standard English?

Reviewer #1: Yes

Reviewer #2: Yes

5. Review Comments to the Author

Reviewer #1: Momosaka and colleagues identified regions of frontal operculum, midcingulate cortex and putamen with lower rCBF in OCD patients using ASL imaging. Moreover, these significant brain regions mostly targeted in the cortico-striatal-thalamic circuit, which provides more evidence of CSTC physiological disruption through ASL modality. It's a very interesting study. The analysis methods appeared to be sound and thus the group differences are very robust. I also have a couple of additional suggestions to improve this manuscript.

Major comments:

1, In the inclusion criteria of OCD patients, the authors did not mention whether the patients have the comorbidity of depression and anxiety symptoms, as comorbidity issue is very common in the clinically psychiatric diagnosis. The author should at least address this point. If yes, whether this comorbidity could have an impact on the results.

2, The authors reported the results of both nonparametric and parametric tests. I think nonparametric results are good enough to illustrate the main findings of this study and more strict to identify the abnormal regions in OCD in this paper. So, I would suggest to remove the results of parametric results.

3, In the introduction, the author mentioned higher signal-to-noise ratio in the pCASL, which could be helpful to sensitively detect the brain abnormalities in OCD. Moreover, they found lower rCBF in the regions of the temporal pole and frontal operculum, which is in or very close to the regions with lower signal-to-noise ratio in the neuroimaging correction. Could the authors test the effect of SNR among these regions showing between-group differences?

4, In the discussion, the authors could add more discussion about the relationship between decreased rCBF and its associated disrupted pathophysiology of OCD. In other words, why does OCD show decreases of rCBF, rather than increased rCBF? What does lower rCBF imply in the links with the symptomatology of OCD?

Minor comments:

1, In the abstract, the authors said the alterations of cerebral perfusion may precede structrual changes in OCD, which does not make sense and is not rigorous. The authors did not have longitudinal results to support this point. It's very difficult to make a conclusion of the order of functional and structural changes for the current results. The authors could just say there is no structural changes in OCD while physiological disruption occurs in OCD patients. Thus, I would suggest to remove this kind of description in the manuscript.

2, In the Methods, the authors did not mention the resample of neuroimaging. Resample to 3*3*3?

3, Even though the authors regressed out total brain volume in the between-group comparisons of rCBF, they should report the results of between-group comparison of total brain volume before they do the VBM analysis. Because sometimes structural changes did not meet the multiple testing in voxel-level analysis, while it could have subtle changes in the global or regional level.

4, In the Methods, the authors run the nonparametric analysis with 5000-time permutation. Why not choose 10000 permutations, which is more commonly used.

Reviewer #2: This paper proposes to identify abnormalities of regional cerebral blood flow (rCBF) in obsessive compulsive disorder (OCD) patients by means of a voxel-based analysis of pseudo continuous arterial spin labeling (pCASL) perfusion images. A study of 23 OCD patients and 64 health controls is conducted by performing a patient recruitment, MRI data acquisition, data preprocessing, Voxel-based morphometry and statistical analysis. The results and discussion are included to show that the rCBF reductions in these regions observed in OCD patients could be associated with the pathophysiology of OCD.

Overall this is an interesting paper that studies an important topic in rCBF analysis of OCD patients. The author novelly proposed to use the entire brain analysis instead of ROI-based methods. Essential research protocols and experiments are set up to study the subjects. The pseudo-continuous ASL (pCASL) perfusion imaging technique is used for MRI to capture and examine the difference in rCBF between patients and healthy controls. In the statistical analysis and discussion, a concrete observation and precise conclusion is summarized with enough discussion about limitation.

From the application and study perspective, this paper proposes a good method in medical field with sufficient experiment details. However, there are several points that can be further improved:

* The data is not clearly explained and published, which may violate the PLOS Data policy. It's suggested to give a location of the data used and made it public available. Specifically, the statistic analysis part doesn't give a detail about data points, there is only limited summary and overview.

* The experiment is a little bit weak to support the effectiveness of proposed approach. It's suggested to compare the results with other baseline approaches (for example, ROI-based approach). Such comparison will make it more clear to readers and highlights the improvement over other existing methods.

Overall this is an interesting paper, I recommend the author to further improve this paper and address above issues to make it better.

6. PLOS authors have the option to publish the peer review history of their article (what does this mean?). If published, this will include your full peer review and any attached files.

Reviewer #1: No

Reviewer #2: No

---

## [Author Response · Author response to Decision Letter 0]

13 Jun 2020

Reviewer #1:

Major comments:

1, In the inclusion criteria of OCD patients, the authors did not mention whether the patients have the comorbidity of depression and anxiety symptoms, as comorbidity issue is very common in the clinically psychiatric diagnosis. The author should at least address this point. If yes, whether this comorbidity could have an impact on the results.

Response: In this study, we excluded OCD patients with a comorbid axis I diagnosis (including mood and anxiety disorders) by using the structured clinical interview for the DSM-IV axis I disorders, patient edition (p.8, lines 99-102). Thus, none of the OCD patients in this study had any comorbid axis I disorders.

2, The authors reported the results of both nonparametric and parametric tests. I think nonparametric results are good enough to illustrate the main findings of this study and more strict to identify the abnormal regions in OCD in this paper. So, I would suggest to remove the results of parametric results.

Response: We have removed the results for the parametric analyses as suggested. All analyses including the comparison of rCBF and gray matter volume between the OCD patients and healthy controls and the regression analysis within the OCD patients used the nonparametric test. The results obtained with the nonparametric test were the same as those obtained with the parametric test.

3, In the introduction, the author mentioned higher signal-to-noise ratio in the pCASL, which could be helpful to sensitively detect the brain abnormalities in OCD. Moreover, they found lower rCBF in the regions of the temporal pole and frontal operculum, which is in or very close to the regions with lower signal-to-noise ratio in the neuroimaging correction. Could the authors test the effect of SNR among these regions showing between-group differences?

Response: In general, ASL using EPI inevitably suffers from susceptibility artifacts in mesial temporal regions due to the proximity of the skull base and air-containing structures (Petcharunpaisan et al., World J Radiol 2010;2). Distortions of raw pCASL images in regions of high magnetic field susceptibility may lead to a low signal-to-noise ratio (SNR) to some extent; we have therefore measured SNRs within the circular region-of-interest (ROI) placed on the right temporal pole and the right frontal operculum on the label images of the pCASL in all subjects. The SNR was also measured in the right cerebellar hemisphere as a reference. 

We calculated the SNRs by using the following formula (Yu et al., BMC Medical Imaging 2018;18):

SNR=0.655 × μ_tissue⁄σ_air 

where μ_tissue is the mean signal intensity within the ROI positioned on each area of the brain, and σ_air is the standard deviation of the signal intensity within the ROI placed on the air.

As a result, the SNR in the right temporal pole was high enough, and there was no major difference in mean SNR between the right temporal pole and the right cerebellar hemisphere (175.8 ± 104.3 vs. 214.0 ± 124.5). 

In addition, as we did not directly analyze raw pCASL images but rather focused on the between-group differences in quantitative rCBF maps, we think that the SNR of the raw pCASL image doesn’t critically affect the final results. However, since the SNR of the right temporal lobe was a little lower in fact, we have added this issue to the limitation section of the Discussion (p. 23-24, line 348-354).

The SNR of the right frontal operculum was also high enough and not lower than that of the right cerebellar hemisphere (358.8 ± 260.0 vs. 214.0 ± 124.5).

4, In the discussion, the authors could add more discussion about the relationship between decreased rCBF and its associated disrupted pathophysiology of OCD. In other words, why does OCD show decreases of rCBF, rather than increased rCBF? What does lower rCBF imply in the links with the symptomatology of OCD?

Response: Considering the neurovascular coupling theory, the reduction in rCBF shown in the present study might reflect functional decline of the corresponding areas. We have clarified this point throughout the Discussion section and more thoroughly discussed the relationships between the functional decline of the areas detected in the present study and the symptomatology of OCD, referencing previous functional MRI studies (p.17-20, 252-299).

Minor comments:

1, In the abstract, the authors said the alterations of cerebral perfusion may precede structrual changes in OCD, which does not make sense and is not rigorous. The authors did not have longitudinal results to support this point. It's very difficult to make a conclusion of the order of functional and structural changes for the current results. The authors could just say there is no structural changes in OCD while physiological disruption occurs in OCD patients. Thus, I would suggest to remove this kind of description in the manuscript.

Response: We have removed the description about the order of functional and structural changes. Instead, we concluded as follows: it was possible that the ASL could sensitively detect physiologically impaired areas without structural abnormalities even in the small sample size (p.21, line 315-317).

2, In the Methods, the authors did not mention the resample of neuroimaging. Resample to 3*3*3?

Response: Thank you for pointing this out. In the present study, the rCBF maps were first resampled to 1�1�1 mm. Specifically, the M0 images are co-registered and resampled to the 3D T1WIs with 1�1�1 mm resolution; then, the same transformations were applied to the raw rCBF maps. After the normalization to the brain template and smoothing, the resolution was finally changed to 2�2�2 mm. We have added a new Fig 2 which illustrates the details of the preprocessing steps applied to the pCASL maps.

3, Even though the authors regressed out total brain volume in the between-group comparisons of rCBF, they should report the results of between-group comparison of total brain volume before they do the VBM analysis. Because sometimes structural changes did not meet the multiple testing in voxel-level analysis, while it could have subtle changes in the global or regional level.

Response: Thank you for this suggestion. We have compared the total brain volume between the OCD patients and healthy controls and confirmed that there was no significant difference. We have added this result to Table 1. In addition, detailed subjects’ information including total brain volume has been added as supplemental data (S1 and S2 Appendices).

4, In the Methods, the authors run the nonparametric analysis with 5000-time permutation. Why not choose 10000 permutations, which is more commonly used.

Response: We have re-calculated all nonparametric analyses with 10,000-time permutations. The main findings are identical. We have replaced the previous 5,000-permuations results with the new results with 10,000-time permutations.

Reviewer #2:

The data is not clearly explained and published, which may violate the PLOS Data policy. It's suggested to give a location of the data used and made it public available. Specifically, the statistic analysis part doesn't give a detail about data points, there is only limited summary and overview.

Response: We have added the subjects' detailed information including individual age, gender, education years, total brain volume, illness duration, and Y-BOCS score as supplemental data (S1 and S2 Appendices). The raw results of the rCBF comparison between the OCD patients and healthy controls acquired by using SnPM13 were also added as S3 Appendix.

Regarding the statistical analysis section, we have added descriptions of the analytical methods, tools and parameters in as much detail as possible (p.13-14, line 179-198).

The experiment is a little bit weak to support the effectiveness of proposed approach. It's suggested to compare the results with other baseline approaches (for example, ROI-based approach). Such comparison will make it more clear to readers and highlights the improvement over other existing methods.

Response: We have performed an ROI-based analysis in addition to the main voxel-based analysis in order to compare the analytical methods. The ROI-based analysis was performed focusing on the areas corresponding to the right putamen, the right frontal operculum, the left midcingulate cortex, and the right temporal pole, which demonstrated significant rCBF reductions in the OCD patients compared to the healthy controls in the voxel-based analysis. The detailed information of the additional ROI-based analysis is provided as Appendix S4.

The ROI-based analysis detected a reduction of the rCBF in the areas corresponding to the right putamen and right temporal pole; however, it failed to detect an rCBF reduction in the right frontal operculum or the left midcingulate cortex (p.16, line 231-237). This might be because the ROIs gathered together large heterogeneous subregions with separate functional roles, resulting in the missing of small foci of decreased rCBF. Thus, it seems that the voxel-based approach has an advantage over the ROI-based approach in detecting smaller areas with rCBF abnormalities. We have added this description to the Discussion (p.22-23, line 327-335).

---

## [Decision Letter · Decision Letter 1]

9 Jul 2020

A voxel-based analysis of cerebral blood flow abnormalities in obsessive-compulsive disorder using pseudo-continuous arterial spin labeling MRI

PONE-D-20-05758R1

Dear Dr. Momosaka,

We’re pleased to inform you that your manuscript has been judged scientifically suitable for publication and will be formally accepted for publication once it meets all outstanding technical requirements.

Kind regards,

Xi Chen

Academic Editor

PLOS ONE

Additional Editor Comments (optional):

Reviewers' comments:

Reviewer's Responses to Questions

**Comments to the Author**

1. If the authors have adequately addressed your comments raised in a previous round of review and you feel that this manuscript is now acceptable for publication, you may indicate that here to bypass the “Comments to the Author” section, enter your conflict of interest statement in the “Confidential to Editor” section, and submit your "Accept" recommendation.

Reviewer #1: All comments have been addressed

Reviewer #2: All comments have been addressed

2. Is the manuscript technically sound, and do the data support the conclusions?

Reviewer #1: Yes

Reviewer #2: Yes

3. Has the statistical analysis been performed appropriately and rigorously? 

Reviewer #1: Yes

Reviewer #2: Yes

4. Have the authors made all data underlying the findings in their manuscript fully available?

Reviewer #1: Yes

Reviewer #2: Yes

5. Is the manuscript presented in an intelligible fashion and written in standard English?

Reviewer #1: Yes

Reviewer #2: Yes

6. Review Comments to the Author

Reviewer #1: All the points that I mentioned have been well addressed. A good paper to identify some interesting regions associated with OCD in ASL.

Reviewer #2: The revised version addressed the major and minor issues in the comments. Specifically, more descriptions about dataset are included in the article with supplement materials. In additional, a ROI-based analysis is added with sufficient details in the statistical analysis and discussion. Some of the key points and questions are also answered and reflected in the corresponding section to be more concise and convincing. I would suggest to accept this paper for publishing.

7. PLOS authors have the option to publish the peer review history of their article (what does this mean?). If published, this will include your full peer review and any attached files.

Reviewer #1: No

Reviewer #2: No

---

## [Editor Report · Acceptance letter]

14 Jul 2020

PONE-D-20-05758R1 

A voxel-based analysis of cerebral blood flow abnormalities in obsessive-compulsive disorder using pseudo-continuous arterial spin labeling MRI 

Dear Dr. Momosaka:

I'm pleased to inform you that your manuscript has been deemed suitable for publication in PLOS ONE. Congratulations! Your manuscript is now with our production department. 

Kind regards, 

on behalf of

Dr. Xi Chen 

Academic Editor

PLOS ONE